# Using the Phecode System to Identify the Preoperative Clinical Phenotypes Associated with Surgical Site Infection in Patients Undergoing Primary Total Knee Arthroplasty: The Sex Differences

**DOI:** 10.3390/jcm11195784

**Published:** 2022-09-29

**Authors:** Ting-Yu Hung, Kuan-Lin Liu, Shu-Hui Wen

**Affiliations:** 1Sports Medicine Center, Hualien Tzu Chi Hospital, Buddhist Tzu Chi Medical Foundation, Hualien 970473, Taiwan; 2Department of Orthopedics, Hualien Tzu Chi Hospital, Buddhist Tzu Chi Medical Foundation, Hualien 970473, Taiwan; 3School of Medicine, Tzu Chi University, Hualien 97004, Taiwan; 4Department of Public Health, College of Medicine, Tzu Chi University, Hualien 97004, Taiwan

**Keywords:** total knee arthroplasty, preoperative clinical phenotypes, Phecodes, surgical site infection

## Abstract

Sex-related differences among comorbid conditions associated with surgical site infection (SSI) after total knee arthroplasty (TKA) are unclear. This population-based cohort study used a novel approach with a Phecode system to evaluate preoperative clinical phenotypes (i.e., comorbid conditions) associated with SSI after TKA and delineate sex-related differences in phenotypes. Using the Taiwan National Health Insurance Research Database (2014–2018), 83,870 patients who underwent TKA were identified. Demographic and SSI data during the 90-day postoperative follow-up were obtained. Comorbidities identified by the International Classification of Diseases within 1 year before TKA were recorded and mapped into Phecodes representing phenotypes. The overall rate of 90-day SSI was 1.3%. In total, 1663 phenotypes were identified among 83,870 patients—1585 and 1458 phenotypes for female (*n* = 62,018) and male (*n* = 21,852) patients, respectively. According to multivariate logistic regression analysis, the SSI odds ratio significantly increased with the presence of each of the 16 phenotypes. Subgroup analysis revealed that the presence of 10 and 4 phenotypes significantly increased SSI risk in both sexes; only one phenotype was common to both sexes. Therefore, comorbid conditions and sex should be considered in preoperative SSI risk evaluation in patients undergoing primary TKA. These findings provide new perspectives on susceptibility, prevention, and treatment in these patients.

## 1. Introduction

Total knee arthroplasty (TKA) is an effective elective surgery performed in elderly patients with knee osteoarthritis. The number of TKA procedures is predicted to grow to 1.26 million in the US by 2030 [1]. Surgical site infection (SSI), with an incidence rate ranging from 1.1% to 4.8%, is a serious complication following TKA, causing considerable morbidity and increasing the economic burden of patients [2,3,4,5,6]. Preoperative evaluation of risk factors, such as patient characteristics and comorbid conditions, allows physicians to provide specific clinical intervention that can reduce the risk of infection following TKA [7,8]. Consequently, investigation of risk factors associated with infection after TKA using registry [9,10,11,12], government health claims [4,13,14,15], and hospital databases [5,6,16,17,18] has gained growing interest [6]. However, there are two important issues that remain to be addressed: First, the various preferred comorbid conditions potentially associated with infection after TKA were evaluated in previous studies. This limitation may be caused by the lack of information available on comorbid conditions in the databases [5,6,9,10,11,12,16,17,18], or by the excessive numbers of International Classification of Diseases (ICD) diagnostic codes that could be used to identify patients’ comorbid conditions [4,13,14,15]. A more comprehensive evaluation of comorbid conditions associated with SSI after TKA is still lacking. Second, several studies have shown that male patients have higher risks of SSI following TKA than female patients [4,9,10,11,12,14,17]. Patients, particularly the elderly, show sex-related differences in the pattern of comorbid conditions that negatively affect the same clinical outcomes [19,20,21]. However, whether there are sex-related differences among comorbid conditions associated with SSI after TKA is a question that has not been studied. Addressing these two important issues could provide insights useful for recognizing differential susceptibility and setting personalized strategies to both prevent and treat SSI in these patients.

Phenome-wide association studies (PheWAS) have been designed to investigate the multiple clinical phenotypes associated with a given genetic variant using ICD diagnostic codes in electronic health record data [22,23]. For PheWAS, a Phecode system has been developed to convert multiple ICD diagnostic codes into a smaller number (<2000) of Phecodes representing various clinical phenotypes, which are mapped to higher-order disease categories [24,25]. The Phecode system, thus, provides an opportunity to comprehensively evaluate the comorbid conditions as risk factors for diseases. Recently, the Phecode system has been found to be successful in evaluating the effect of comorbid conditions on certain disease outcomes [26,27,28].

There were two main objectives of this study. The Phecode system was used to comprehensively evaluate the preoperative clinical phenotypes associated with SSI in patients undergoing primary TKA, and to delineate differences in sex in these clinical phenotypes. To achieve this goal, we extracted all relevant ICD-9-CM and ICD-10-CM codes from the Taiwan National Health Insurance Research Database (NHIRD) and mapped these ICD diagnostic codes into Phecodes for subsequent association analysis.

## 2. Materials and Methods

### 2.1. Data Sources

This population-based cohort study was conducted using the NHIRD from 2014 to 2018. The National Health Insurance program provides the entire population in Taiwan with universal health insurance, and covers 99% of their healthcare needs and medical services. The NHIRD has detailed medical records for each patient, including information about outpatient visits, patient admissions, surgical procedures, medication prescriptions, and health providers [29]; basic demographic variables of sex and age; and disease diagnosis codes from both outpatient and inpatient records. In this study, the disease diagnostic codes collected were based on coding systems, including the ICD-9-CM and ICD-10-CM codes. Their validity and accuracy have been demonstrated in previous studies [29]. This study was approved by the research ethics committee. The NHIRD encrypts patients’ personal information, including personal identification numbers, birth date, and names to protect privacy [30]. The requirement for informed consent was waived according to the institutional guidelines for this retrospective study.

### 2.2. Study Samples

Adult patients who were aged more than 18 years and who had undergone primary unilateral TKA (ICD-9-CM procedure code: 81.54, ICD-10-PCS codes: 0SRC, 0SRD, 0SRT, 0SRU, 0SRV, and 0SRW) between 1 January 2015 and 30 September 2018 were identified. Patients were excluded if they had (1) indications that the need for TKA was due to trauma or traffic accident, as indicated by diagnostic codes, (2) incomplete follow-up, (3) missing data, or (4) existing infection before TKA. After the exclusions, the remaining 83,870 patients were included in the study cohort (Figure 1).

### 2.3. Demographic Data and Primary Outcome

Data on basic demographic characteristics, including age, sex, and year of TKA, were collected. The primary outcome was SSI, which refers to an infection that occurs within 90 days of surgery. SSI was identified according to the occurrence of diagnostic codes or procedural codes within 90 days after TKA. The diagnostic codes included intra-articular infection, postoperative infection, pyogenic arthritis, and osteomyelitis. The procedural codes representing surgical procedures for TKA infection included debridement, osteomyelitis debridement, acute septic joint arthrotomy, and prosthesis removal. These identification strategies were suggested by the program of healthcare-associated infection indicators conducted by the Taiwan Centers for Disease Control [31].

### 2.4. Phecodes for Evaluation of Preoperative Clinical Phenotypes

The Phecode system contains 1866 hierarchical phenotype codes that were mapped from the ICD-9-CM or ICD-10-CM codes to various disease categories [25]. The mapping rules of Phecodes Version 1.2 were followed (http://phewascatalog.org, accessed on 21 March 2022). To identify preoperative comorbidities, ICD diagnosis codes were extracted within 1 year before the TKA operation date. To reduce coding errors, comorbidity was defined as the presence of situations when ICD diagnostic codes appeared at least twice in the outpatient records, or once in the inpatient data for any study patient undergoing TKA. A total of 13,014 diagnostic codes pertaining to the study patients were extracted and transferred into the Phecode system [24,25]. Therefore, 1663 Phecodes representing preoperative clinical phenotypes were identified among the study patients, with 1585 and 1458 Phecodes for female and male patients, respectively.

### 2.5. Statistical Analysis

Continuous variables, compared using an independent two sample *t*-test, are presented as mean ± standard deviation (SD). Categorical variables, compared using the Chi-square test, are presented as frequencies and percentages. Logistic regression analysis was used to evaluate the association between clinical phenotypes and SSI while adjusting for potential confounders. The adjusted odds ratios (ORs) and 95% confidence intervals (CIs) were obtained. In the association analysis of certain clinical phenotypes with a low frequency of SSI, the Firth’s penalised maximum likelihood logistic regression was performed to reduce estimation bias [32]. The PheWAS R package (version 0.99.5-5, Robert Carroll, Nashville, TN, USA) and the logistf R package (version 1.24, Georg Heinze, Vienna, Austria) were used to analyze the association between clinical phenotypes and SSI. The Bonferroni correction threshold was used to account for multiple comparisons; all *p*-values below the significance level of 0.1 divided by the total number of Phecodes analysed were significant. We set the significance threshold at a *p*-value less than $\frac{0.1}{1663}=6.013\times 10^{-5},\frac{0.1}{1585}=6.309\times 10^{-5},\;\mathrm{and}\;\frac{0.1}{1458}=6.859\times 10^{-5}$ after Bonferroni correction for all samples, for the female group, and for the male group, respectively. This is a less conservative significance level, as 0.1 was set to compensate for the reduction in the power of the multiplicity test because of the large number of clinical phenotypes and reduced sample size in the subgroup analysis. When two-tailed *p* values were <0.05, they were considered statistically significant in the case of group comparisons, and when *p* values were <0.1/number of tests, they were considered statistically significant in the case of the Bonferroni correction method. All statistical analysis was performed using the R software program, version 4.0.5 (R Development Core Team, Vienna, Austria; https://www.r-project.org/, accessed on 21 March 2022).

## 3. Results

### 3.1. Characteristics of Participants

Among the 83,870 eligible patients who underwent primary TKA, there were 62,018 female patients (73.9%) and 21,852 male patients (26.1%) (Figure 1). Table 1 shows the distribution of basic demographic characteristics among the study patients and sex subgroups. Across all samples, the average age was 70.5 ± 8.4 years, and there was a slight difference in age between the sex subgroups. The overall SSI rate during the 90-day follow-up period was 1.3%. The rate of SSI was significantly higher in male patients than in female patients (1.7% vs. 1.2%, *p* < 0.001). The distribution of surgical years of TKA in male patients did not significantly differ from that in female patients (*p* = 0.067).

### 3.2. Associations between Preoperative Clinical Phenotypes and Postoperative Surgical Site Infection

After analyzing the patients’ comorbid conditions within a year before TKA, 1663 Phecodes were identified: 1585 for female patients and 1458 for male patients. These Phecodes were used to analyze the relationship between preoperative clinical phenotypes and 90-day SSI, and the results are presented in the Manhattan plots (Figure 2). As shown, 16, 10, and 4 clinical phenotypes were identified as having a significant association with SSI (indicated by dots above the red horizontal line) among the study patients and in the two sex subgroups. Details of these clinical phenotypes are presented in Table 1. Female patients had a significantly lower incidence in the majority (15 of 21) of these clinical phenotypes than male patients. The preoperative clinical phenotypes identified as risk factors for SSI included cellulitis of the extremities, open wounds of the extremities, atrial flutter, pyogenic arthritis, contusion, unspecified anemias, septicemia, and back pain (Table 1, Figure 2 and Figure 3). However, there were three (sleep-related leg cramps, pleurisy–pleural effusion, and cholesteatoma) and two clinical phenotypes (sepsis and fracture of unspecified part of femur) in the female and male subgroups, respectively, that were not identified in the analysis of all samples (Figure 3). There was only one single clinical phenotype (cellulitis and abscess of arm/hand) that was common to both sex subgroups (Figure 3).

### 3.3. Preoperative Clinical Phenotypes That Increased the Risk of Postoperative Surgical Site Infection

The relationship between preoperative clinical phenotypes and 90-day SSI after TKA was further evaluated using multivariate logistic analyses, and the results are reported in Table 2. The presence of each of these clinical phenotypes significantly increased the OR of SSI following TKA after adjusting for potential confounders. Forest plots showed an increased risk of developing SSI in patients who underwent TKA due to the presence of these clinical phenotypes (Figure 4). The clinical phenotypes of the female subgroup were superficial cellulitis and abscess, other local infections of the skin and subcutaneous tissue, cellulitis and abscess of the leg except the foot, cellulitis and abscess of the arm/hand, atopic/contact dermatitis due to other or unspecified causes, other anemias, sleep-related leg cramps, pleurisy–pleural effusion, cholesteatoma, and back pain. In the male subgroup, cellulitis and abscess of the arm/hand, sepsis, fracture of the unspecified part of the femur, and atrial fibrillation were observed.

## 4. Discussion

In this nationwide population-based cohort study, a novel approach was considered, using the Phecode system to comprehensively evaluate the preoperative clinical phenotypes (comorbid conditions) associated with SSI in TKA patients. The first major finding of this study was the presence of 16 preoperative clinical phenotypes that increased the risk of SSI among patients undergoing primary TKA. A second major finding was that sex-related differences in the types of preoperative clinical phenotypes were risk factors for SSI. This study is among the first few investigations [26,27,28] to use the Phecode system to evaluate the effect of comorbid conditions on certain disease outcomes.

In this cohort study, the rate of 90-day SSI was 1.3%, which is within the range previously reported [3,9]. Preoperative identification of comorbid conditions allows physicians to optimize therapeutic strategies and reduce the risk of SSI following TKA [7,8] Several comorbid conditions have been reported to significantly increase predisposition for SSI in patients undergoing TKA, most notably diabetes [4,7,8,12,13,16,18], obesity [4,5,7,8,9,12,13,16,18], metastatic cancer [4,13,15], chronic pulmonary disorders [4,13], cardiovascular diseases [4,6,7,13], anemia [4,7,8,13], rheumatoid arthritis [4,7,8,13,14], mental disorders [4,7,13,17], renal diseases [4,7,8,13], and malnutrition [4,7,8,16]. However, varying preferred comorbid conditions were selected in these previous studies. For example, one study selected 34 comorbid conditions [4], whereas another study only selected three comorbid conditions [6]. Even when a large number of comorbid conditions were assessed, a more comprehensive evaluation was still lacking. This limitation may be due to the limited availability of information in the databases [5,6,9,10,11,12,16,17,18] or the excessive numbers of ICD diagnostic codes for the assessment of comorbid conditions [4,13,14,15]. The novel approach of this study using the Phecode system provides a means of overcoming this limitation.

The findings of the analysis of preoperative clinical phenotypes as risk factors among the study patients agree with previous observations. For example, findings on the clinical phenotypes of unspecified anemias, atrial flutter, atrial fibrillation, and pyogenic arthritis have been previously reported [4,7,8,13]. Findings regarding several clinical phenotypes in the dermatologic category involving prior cellulitis or infection have also been reported previously [17]. Several clinical phenotypes in the injury category involving open wounds or contusions could be regarded as risk factors for prior open surgical procedures [9,16]. However, we additionally found that clinical phenotypes of spondylosis with myelopathy, septicemia, and back pain were risk factors for SSI. Some of the comorbid conditions previously reported as risk factors for SSI, such as diabetes, metastatic cancer, rheumatoid arthritis, mental disorders, and renal diseases, were not supported in this study, and this result was also reported by several previous investigations [5,10,18], particularly in the Chinese population [5,18]. A recent meta-analysis [33] included 14 studies to calculate the risk of developing an infection in diabetic patients compared to non-diabetic patients following primary TKA. Among these included studies, six reported that diabetes increased the risk ratio of infection, while the other eight showed a null effect of diabetes. Although the authors [33] demonstrated a significant indication of diabetes as a risk factor, they concluded that the risk is much lower than implied by previously reported data, suggesting that other factors play a larger role in infection in this patient population.

In this study, male patients were found to have a significantly higher rate of SSI than female patients, a finding that is in good agreement with previous observations [4,9,10,11,12,14,17]. However, in previous studies, data from both sexes were pooled together to investigate risk factors for infection. Sex-related differences in the pattern of comorbid conditions affecting the risk of SSI remain unknown. It has been reported that there are sex-related differences in the patterns of comorbid conditions associated with mortality [19] and frailty [20] in elderly people, as well as in patients with heart failure [21]. The results showed that there was a sex-related difference in the types of preoperative clinical phenotypes as risk factors; only one comorbid condition (cellulitis and abscess of the arm and/or hand) was common to both sex subgroups. In addition, the results showed that female patients had more comorbid conditions that might increase the risk of SSI than male patients, a finding that seems to be paradoxical when considering the higher rate of SSI in males. This paradox can be explained by the sex-based immunological differences that contribute to the dissimilar susceptibility to infection [34,35]. It should be noted that of 16 clinical phenotypes, eight were no longer risk factors in the subgroup analysis. This may be because of the reduced sample size after subgrouping. In addition, there were three and two clinical phenotypes in the female and male subgroups, respectively, which were not identified in the analysis of all the study patients. This might be a consequence of homogenizing the study samples into a single sex group.

The strength of this work is that the feasibility of using the Phecode system in analyzing real-world data was demonstrated comprehensively, evaluating the comorbid conditions associated with a specific disease in a large patient cohort. However, the current study has some limitations. First, the NHIRD, which is a large national administrative database, was used. The method of disease diagnosis coding may create potential errors in identifying comorbid conditions, although this is not dissimilar to the risk when using any other claims database. Second, the NHIRD does not include detailed patient and surgical factors, which may limit our analysis and findings.

## 5. Conclusions

The approach of using the Phecode system is novel and clinically relevant for evaluating the effects of preoperative clinical phenotypes on the risk of SSI in patients undergoing primary TKA. The results of this study show that previously unidentified comorbid conditions should be considered for the preoperative evaluation of risk factors for SSI in patients undergoing TKA. In addition, our results show that sex as an important variable in this preoperative evaluation. Our findings may provide a new perspective for the differential susceptibility, prevention, and treatment of SSI in these patients.

## Figures and Tables

**Figure 1 jcm-11-05784-f001:**
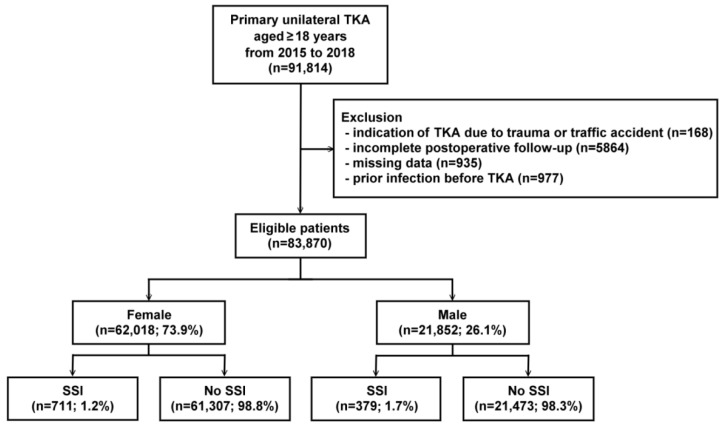
Flow chart of the study. TKA, total knee arthroplasty; SSI, surgical site infection.

**Figure 2 jcm-11-05784-f002:**
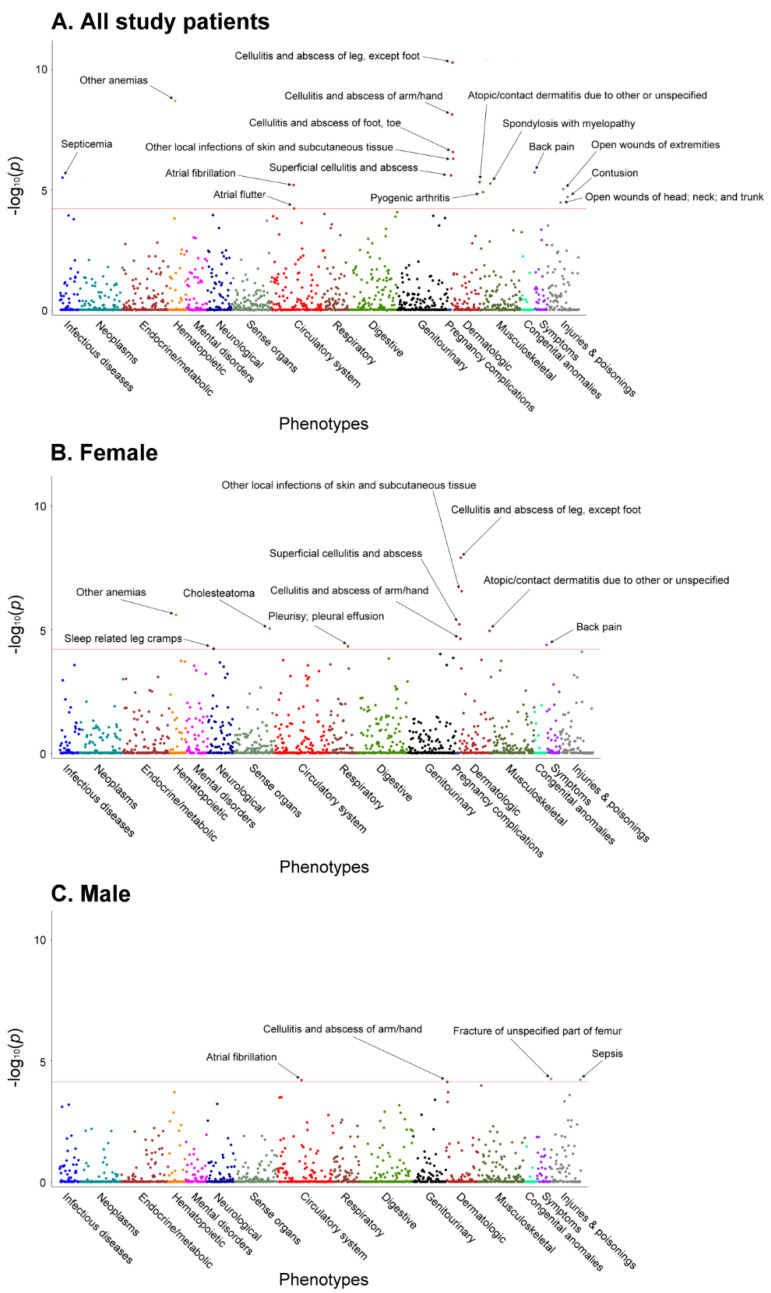
Manhattan plot showing the relationship between preoperative clinical phenotypes and surgical site infection amongst study patients (**A**), and in two sex subgroups (**B**,**C**). A total of 1663 clinical phenotypes mapped to their corresponding disease categories are represented in different colors along the x-axis. Meanwhile, the y-axis represents the −log_10_ (*p*-value) of the association. The red horizontal line represents the significance threshold of association adjusted by Bonferroni correction (*p* < 0.1/number of tests). In the analysis amongst study patients, the model was adjusted by age, sex and surgical year of TKA, whereas in the analysis of each sex group, the model was adjusted by age and surgical year of TKA.

**Figure 3 jcm-11-05784-f003:**
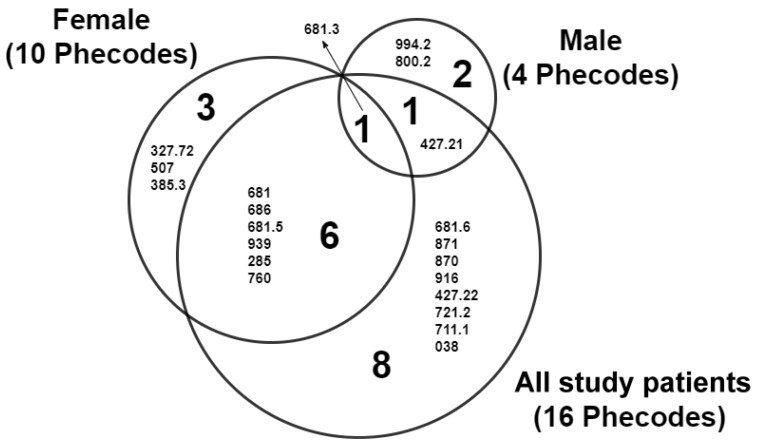
Venn diagram illustrating the preoperative clinical phenotypes that were significantly associated with SSI amongst study patients and in two sex subgroups. Small numbers in the diagram are Phecodes representing different clinical phenotypes. Note that 16, 10, and 4 clinical phenotypes were identified in the study patient subgroup, the female subgroup, and the male subgroup, respectively.

**Figure 4 jcm-11-05784-f004:**
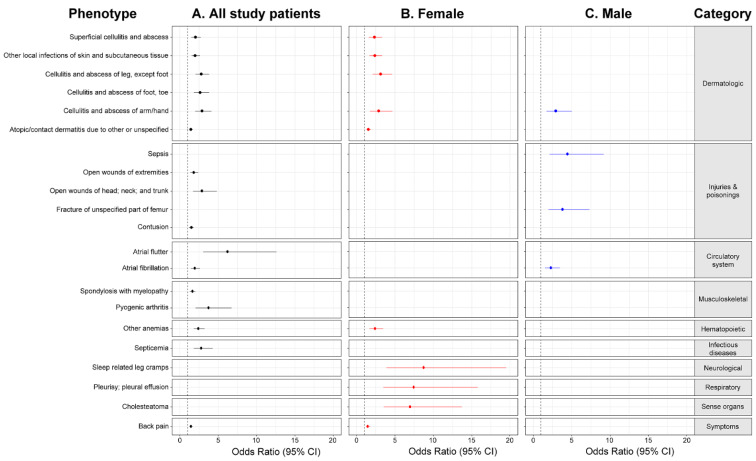
Forest plots of odds ratios and 95% confidence intervals (CI) showing that the presence of preoperative clinical phenotypes increased the risk of contracting SSI amongst study patients (**A**), and in the two sex subgroups (**B**,**C**). Clinical phenotypes (leftmost) were mapped to their corresponding disease categories (rightmost). The vertical dotted line indicates an odds ratio of 1. Note that 16, 10, and 4 clinical phenotypes were analyzed amongst the study patients subgroup, the female subgroup, and the male subgroup, respectively.

**Table 1 jcm-11-05784-t001:** Demographic characteristics and preoperative clinical phenotypes in all study patients and in two sex subgroups.

			All	Female	Male	*p**-*Value
**Demographic characteristics**		Sample size	83,870	62,018 (73.9)	21,852 (26.1)	
		Age (years)	70.5 ± 8.4	70.4 ± 8.0	70.7 ± 9.4	<0.001
		Surgical year of TKA				0.067
		2015	21,904 (26.1)	16,119 (26.0)	5785 (26.5)	
		2016	21,870 (26.1)	16,229 (26.2)	5641 (25.8)	
		2017	22,764 (27.1)	16,941 (27.3)	5823 (26.6)	
		2018	17,332 (20.7)	12,729 (20.5)	4603 (21.1)	
		Surgical site infection	1090 (1.3)	711 (1.2)	379 (1.7)	<0.001
**Disease category**	**Phecode**	**Clinical phenotypes**				
Dermatologic						
	681	^a^ Superficial cellulitis and abscess	1795 (2.3)	1203 (2.1)	592 (2.9)	<0.001
	686	^a^ Other local infections of skin and subcutaneous tissue	2262 (2.8)	1540 (2.6)	722 (3.5)	<0.001
	681.5	^a^ Cellulitis and abscess of leg, except foot	1339 (1.7)	871 (1.5)	468 (2.3)	<0.001
	681.6	Cellulitis and abscess of foot, toe	1046 (1.3)	650 (1.1)	396 (2.0)	<0.001
	681.3	^a,b^ Cellulitis and abscess of arm/hand	1077 (1.4)	686 (1.2)	391 (1.9)	<0.001
	939	^a^ Atopic/contact dermatitis due to other or unspecified	9985 (12.3)	6949 (11.5)	3036 (14.3)	<0.001
Injuries & poisonings						
	994.2	^b^ Sepsis	469 (0.6)	324 (0.5)	145 (0.7)	0.019
	871	Open wounds of extremities	2352 (2.9)	1468 (2.4)	884 (4.2)	<0.001
	870	Open wounds of head; neck; and trunk	378 (0.5)	232 (0.4)	146 (0.7)	<0.001
	800.2	^b^ Fracture of unspecified part of femur	652 (0.8)	481 (0.8)	171 (0.8)	0.789
	916	Contusion	6240 (7.4)	4665 (7.5)	1575 (7.2)	0.132
Circulatory system						
	427.22	Atrial flutter	114 (0.1)	62 (0.1)	52 (0.3)	<0.001
	427.21	^b^ Atrial fibrillation	1892 (2.4)	1203 (2.1)	689 (3.4)	<0.001
Musculoskeletal						
	721.2	Spondylosis with myelopathy	4620 (7.3)	3538 (7.6)	1082 (6.3)	<0.001
	711.1	Pyogenic arthritis	287 (0.6)	143 (0.4)	144 (1.2)	<0.001
Hematopoietic						
	285	^a^ Other anemias	1635 (2.0)	1152 (1.9)	483 (2.2)	0.001
Infectious diseases						
	038	Septicemia	604 (0.7)	398 (0.7)	206 (1.0)	<0.001
Neurological						
	327.72	^a^ Sleep-related leg cramps	120 (0.2)	89 (0.2)	31 (0.2)	0.844
Respiratory						
	507	^a^ Pleurisy—pleural effusion	180 (0.2)	94 (0.2)	86 (0.4)	<0.001
Sense organs						
	385.3	^a^ Cholesteatoma	134 (0.2)	101 (0.2)	33 (0.2)	0.78
Symptoms						
	760	^a^ Back pain	11,702 (14.0)	8874 (14.3)	2828 (12.9)	<0.001

A total of 1663 Phecodes representing various preoperative clinical phenotypes were mapped to their corresponding disease categories and were used for the analyses. Only clinical phenotypes that were significantly associated with surgical site infection are presented in this table. Data on age are presented as mean ± SD, and were compared using independent two-sample *t*-test. Other data are presented as frequency (%), and were compared using chi-square test. A *p*-value < 0.05 was considered statistically significant between the two sex groups. TKA, total knee arthroplasty; ^a^ specific to females; ^b^ specific to males.

**Table 2 jcm-11-05784-t002:** Multivariate analysis of associations between preoperative clinical phenotypes and surgical site infection in all study patients and in two sex subgroups.

Disease Category	Clinical Phenotypes	All			Female			Male		
		No. of SSI	No. of Controls	OR	No. of SSI	No. of Controls	OR	No. of SSI	No. of Controls	OR
**Dermatologic**										
	^a^ Superficial cellulitis and abscess	992	77,977	2.05 *	646	58,000	2.33 *	346	19,977	1.66
	^a^ Other local infections of skin and subcutaneous tissue	1001	78,435	2.01 *	653	58,330	2.39 *	348	20,105	1.51
	^a^ Cellulitis and abscess of leg, except foot	989	77,524	2.81 *	642	57,672	3.13 *	347	19,852	2.43
	Cellulitis and abscess of foot, toe	975	77,245	2.65 *	630	57,463	2.58	345	19,782	2.75
	^a,b^ Cellulitis and abscess of arm/hand	977	77,274	2.91 *	632	57,497	2.89 *	345	19,777	2.95 *
	^a^ Atopic/contact dermatitis due to other or unspecified	1050	80,409	1.46 *	684	59,549	1.57 *	366	20,860	1.30
**Injuries & poisonings**										
	^b^ Sepsis	1090	82,766	2.32	711	61,300	1.02	379	21,466	4.44 *
	Open wounds of extremities	1042	80,330	1.85 *	692	59,767	1.86	350	20,563	1.86
	Open wounds of head; neck; and trunk	1001	78,397	2.93 *	670	58,553	3.21	331	19,844	2.64
	^b^ Fracture of unspecified part of femur	980	76,783	2.40	627	56,322	1.63	353	20,461	3.83 *
	Contusion	1090	82,780	1.52 *	711	61,307	1.60	379	21,473	1.37
**Circulatory system**										
	Atrial flutter	991	75,263	6.22 *	650	55,786	9.31	341	19,477	4.36
	^b^ Atrial fibrillation	1032	77,000	1.96 *	668	56,909	1.68	364	20,091	2.31 *
**Musculoskeletal**										
	Spondylosis with myelopathy	826	62,753	1.66 *	525	45,767	1.66	301	16,986	1.68
	Pyogenic arthritis	677	44,779	3.74 *	440	32,709	2.91	237	12,070	4.37
Hematopoietic										
	^a^ Other anemias	1068	81,501	2.43 *	697	60,395	2.41 *	371	21,106	2.46
**Infectious diseases**										
	Septicemia	1065	80,896	2.79 *	694	59,972	2.63	371	20,924	3.03
**Neurological**										
	^a^ Sleep-related leg cramps	913	70,136	6.49	577	51,257	8.72 *	336	18,879	3.48
**Respiratory**										
	^a^ Pleurisy; pleural effusion	1080	82,305	3.91	705	61,057	7.44 *	375	21,248	1.74
**Sense organs**										
	^a^ Cholesteatoma	1069	81,498	4.73	695	60,358	6.99 *	374	21,140	0.74
**Symptoms**										
	^a^ Back pain	1090	82,780	1.45 *	711	61,307	1.48 *	379	21,473	1.40

A total of 1663 Phecodes representing preoperative clinical phenotypes were mapped to their corresponding disease categories and were used for the analyses. Only clinical phenotypes that were significantly associated with SSI are presented in this table. The Bonferroni corrections were used to adjust *p* values at an alpha-level of 0.1. * statistical significance (*p* < 0.1/number of tests); ^a^ specific to females; ^b^ specific to males; SSI, surgical site infection; ORs, odds ratios obtained from multivariate logistic regression adjusted by potential confounders.

## Data Availability

The data for this study were obtained from the National Health Insurance Research Database provided by the Bureau of National Health Insurance, Department of Health, and managed by the Health and Welfare Statistics Application Center, Ministry of Health and Welfare. The interpretations and conclusions contained herein do not represent those of the Bureau of National Health Insurance, the Department of Health, or the Ministry of Health and Welfare.

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
