# Peer review of "Using the Phecode System to Identify the Preoperative Clinical Phenotypes Associated with Surgical Site Infection in Patients Undergoing Primary Total Knee Arthroplasty: The Sex Differences"

_jcm, 2022, doi:10.3390/jcm11195784_

Round 1

Reviewer 1 Report

Dear Authors, all sections are well balanced but i suggest to improve the references citing the following article:

-An unusual complication after infected total knee arthroplasty

Solarino G. et al.

Joints Open AccessVolume 6, Issue 4, Pages 241 - 2452018

By this article the reader could underline the importance of prevention of prosthetic infection 

Reviewer 2 Report

The authors have presented an interesting study where they have presented some new perspectives on evaluating surgical site infection information using a phecode system on a sex based differences. They study is well-written and I only have some questions/comments.

1) How were the 16 preoperative clinical phenotypes decided that increased the risk of SSI? Can the authors add a little more information on how they decided/conclude on the 16 clinical phenotypes?

2) For the patient population under consideration for the study, were there any patients (either male or female or both) whose age group was outside of the average age mentioned? If yes, were they considered for the study? 

3) The authors can present a improved Figure 2 with better clarity especially the phenotypes above the threshold line are extremely difficult to read. 

Reviewer 3 Report

The work “Using Phecode system to identify the preoperative clinical phenotypes associated with surgical site infection in patients undergoing primary total knee arthroplasty: the sex differences” has a significant number of patients what allows authors to perform confidant data analysis.

However, some topics must be clarified by authors.

In the abstract authors should avoid abbreviations for better understanding.

Line 80-81

It is not enough to say that the work had the approval of the ethics committee. As the data comes from the National Health Insurance, it is necessary to clarify the protection of the personal data of the patients consulted. Authors must prove that this type of confidentiality and data protection exists.

Figure 2 is too small and doesn’t allow a correct reading.

One of the most frequent comorbidities for infections is metabolic diseases, such as diabetes. In the discussion the authors mention this pathology, but it is not clear why diabetes was not considered in the study. Therefore, the authors are invited to review the discussion to clarify this point.
